# Nutritional Support for Necrotizing Soft Tissue Infection Patients: From ICU to Outpatient Care

**DOI:** 10.3390/jcm14093167

**Published:** 2025-05-03

**Authors:** Eftikhar A. Akam, Stacy L. Pelekhaty, Caitlin P. Knisley, Michael G. Ley, Noah V. Loran, Eric J. Ley

**Affiliations:** R Adams Cowley Shock Trauma Center, University of Maryland, Baltimore, MD 21201, USA

**Keywords:** necrotizing soft tissue infection (NSTI), nutrition, phases of recovery, energy expenditure, protein needs

## Abstract

Although nutrition recommendations for patients with necrotizing soft tissue infections (NSTIs) often parallel those for patients with burn injuries, differences in the metabolic response to stress indicate that NSTIs require a unique approach. The sepsis and wound management associated with NSTIs trigger a metabolic response, driven by inflammatory and neuroendocrine changes, that leads to high circulating levels of cortisol, catecholamines, insulin, and pro-inflammatory cytokines. This metabolic response follows four phases of recovery (Early Acute; Late Acute; Persistent Inflammation, Immunosuppression, and Catabolism Syndrome; Recovery) that require a thoughtful approach to nutrition by risk screening, malnutrition assessment, and micronutrient deficiency assessment. Close monitoring of energy expenditure and protein needs is required for appropriate nutrition management. Nutrition intake after transfer from the intensive care unit and hospital discharge is often inadequate. Ongoing monitoring of nutrition intake at all outpatient follow-up appointments is necessary, regardless of the route of delivery, until the nutrition status stabilizes and any nutritional decline experienced during hospitalization has been corrected.

## 1. Introduction

Necrotizing soft tissue infections (NSTIs) are highly aggressive infections with an approximate incidence in the United States of 0.4 per 1000, although the actual incidence is unknown due to variability in reporting [1]. NSTIs are often referred to burn centers due to these injuries’ similarities, including their rapid tissue destruction and wound care needs. However, Endorf et al. were unable to show a mortality benefit with the treatment of NSTIs at burn centers [2]. This is in part related to differences in wound characteristics but also likely due to variations in the metabolic response between the populations. While patients with NSTIs and burn injuries both exhibit an increased metabolic rate because of the metabolic response to stress, only patients with burns will experience mitochondrial decoupling for thermogenesis [3,4,5].

The pathogenesis of NSTIs is secondary to bacterial invasion, the release of virulence factors, and systemic inflammatory responses [6]. Ultimately, this leads to sepsis, coagulopathy, and multisystem organ failure. Risk factors for the development of NSTIs include diabetes mellitus, obesity, malnutrition, peripheral vascular disease, and immunosuppression [7]. Antimicrobial therapy, early and aggressive surgical debridement, meticulous wound care, and nutritional optimization are key to managing NSTIs.

NSTIs are generally categorized into four types based on the microorganism causing the infection. Type I NSTIs are the most common and tend to be polymicrobial, with diabetes being a significant risk factor. Type II infections are monomicrobial and are often caused by streptococcal and staphylococcal species. Type III and Type IV NSTIs are less common and are caused by vibrio and fungal species, respectively (Table 1).

The importance of nutrition in the management of NSTI patients cannot be understated. This chapter focuses on the essential role of nutrition after the diagnosis of an NSTI, from the intensive care unit (ICU) to outpatient recovery, emphasizing its importance in supporting immune function and promoting wound healing.

## 2. Metabolic Response to Stress and Injury

In healthy individuals, the body’s energy needs are met through various metabolic pathways. The primary energy sources—carbohydrates, lipids, and proteins—are metabolized to produce adenosine triphosphate (ATP), the energy currency of the body. Carbohydrates are the body’s main source of energy, while lipids serve as dense energy reserves, and proteins are utilized for energy metabolism only in states of severe deprivation or macronutrient imbalance.

NSTIs trigger a stress response driven by inflammatory and neuroendocrine changes, characterized by high circulating levels of cortisol, catecholamines, insulin, and pro-inflammatory cytokines [8,9]. Historically, there are three key phases of stress metabolism (Table 2). However, a fourth potential phase, Persistent Inflammation, Immunosuppression, and Catabolism Syndrome (PICS), occurs in 40–60% of critical care survivors [10,11].

The current understanding of the metabolic response is as follows:**Early Acute Phase**: This phase is marked by a hypometabolic state with decreased energy consumption, diminished mitochondrial capacity, and hemodynamic instability. In patients presenting in septic shock, as many patients with NSTIs do, the Early Acute Phase may be more pronounced [12].**Late Acute Phase**: This catabolic, hypermetabolic phase begins 2–7 days after injury and may last for weeks. It is characterized by increased energy expenditure, muscle proteolysis, and rapid tissue breakdown. Isocaloric nutrition support with an adequate protein content may blunt losses, but resolution of the underlying stress state is required to return to metabolic homeostasis.**Persistent Inflammation, Immunosuppression, and Catabolism Syndrome [10]**: Metabolic PICS, first described in 2012, is characterized by ongoing, smoldering inflammation, with infectious recidivism, anabolic resistance, exercise intolerance, and skeletal muscle catabolism due to ongoing inflammation. Necrosis and apoptosis of renal endothelium and skeletal muscle release damage-associated molecular patterns (DAMPs), which increase the risk of developing metabolic PICS. Additionally, chronic critical illness and repeated mobilizations of stress metabolism increase the likelihood of progressing into metabolic PICS rather than returning to homeostatic conditions.**Recovery or Convalescent Phase**: Eventually, the body returns to metabolic homeostasis as the inflammatory response subsides. This creates favorable conditions for anabolism and nutritional rehabilitation if malnutrition developed during hospitalization.

Unlike starvation, these metabolic derangements cannot be reversed with nutrition interventions. A carefully managed balance between providing necessary nutrition and avoiding overfeeding is required, as either underfeeding or overfeeding could worsen immune function and recovery. The use of screening to assess nutrition needs facilitates this balance.

## 3. Nutrition Risk Screening and Malnutrition Assessment

### 3.1. Nutritional Risk Screening

The purpose of nutrition screening is to identify patients who are at risk of having or developing malnutrition and who may require targeted nutrition interventions during hospitalization. There are many validated clinical tools that can assess the nutritional risk of critically ill patients, and the optimal tool depends on the resources available. The Academy of Nutrition and Dietetics conducted a comprehensive review of screening tools and recommends the use of the Malnutrition Screening Tool (MST) as an easily completed screening method with high validity for identifying individuals at risk of malnutrition across a range of ages and care settings [13]. The American Society for Parenteral and Enteral Nutrition (ASPEN) 2016 guidelines recommend the Nutrition Risk if Critically Ill (NUTRIC) or Nutritional Risk Screening 2002 (NRS 2002) tools related to their specific development for use in intensive care [14]. The European Society for Clinical Nutrition and Metabolism (ESPEN) recommends that all patients remaining in the intensive care unit for more than 48 h be considered at nutrition risk [15]. In patients who are not identified as being at nutrition risk upon presentation to the hospital, nutrition screening should be conducted periodically throughout hospitalization to ensure that changes in the risk level appropriately trigger a nutrition consult. In patients with NSTIs, failure of the wound to progress as anticipated warrants a consult to a nutrition expert for a careful evaluation of nutritional adequacy, including macronutrient and micronutrient intake.

### 3.2. Malnutrition Assessment

Diagnosing malnutrition requires a holistic assessment performed by a trained professional (Table 3). Subjective Global Assessment (SGA) is considered the gold standard for malnutrition assessment and incorporates nutrition intake and weight history, functional status, gastrointestinal symptoms, muscle wasting, and edema [16]. The Academy of Nutrition and Dietetics and ASPEN Indicators of Malnutrition (AAIM) framework was published in 2011 and incorporates adipose tissue wasting and considerations of the etiology of malnutrition in addition to assessment parameters described by SGA [17]. The most recently published framework is the Global Leadership Initiative on Malnutrition (GLIM) criteria [18]. Similarly to AAIM, GLIM includes both etiologic and phenotypic criteria, which must be temporally related [18]. However, GLIM is designed for use across resource-rich and resource-poor environments by health professionals from varied clinical backgrounds [18]. The use of visceral proteins in acute illness is no longer recommended to assess nutrition status or determine the response to nutrition therapy due to the hepatic reprioritization induced by inflammation [19].

### 3.3. Micronutrient Deficiencies

Stress metabolism, inflammation, and wound healing increase the demand on multiple micronutrients; however, prophylactic supplementation is generally not recommended. The most cited micronutrients in wound healing are vitamin C, vitamin A, and zinc [20,21,22]. An assessment of micronutrient deficiency requires careful consideration of biochemical, physical exam, and nutrition history findings as each can be skewed during critical illness. Vitamin C is required for the prolination of collagen during wound healing [22]. Deficiency has been described in critically ill adults, with doses exceeding standard recommendations needed to correct serum levels [23]. However, supratherapeutic doses are not associated with short- or long-term clinical benefit, potentially related to the pro-inflammatory effects [24,25]. Vitamin A stimulates collagen synthesis and epithelial growth; however, over-supplementation is associated with gastrointestinal symptoms, loss of appetite, and liver damage [20,26]. Finally, zinc deficiency is associated with delayed wound closure and weaker scar formation [21]. Conversely, zinc toxicity impedes wound healing, and unnecessary enteral supplementation increases the risk of iron and copper deficiencies and gastrointestinal complications, which may lead to a decline in nutrition intake and nutrition status [27,28].

## 4. Assessment of Nutritional and Metabolic Needs

### 4.1. Initial Assessment

The nutritional assessment of NSTI patients must be comprehensive and take into consideration their medical history, physical examination findings, and laboratory data (Table 4). Important information to elicit includes changes in weight and appetite and evaluations for signs of malnutrition (e.g., poor wound healing or loss of muscle mass). Laboratory tests such as those to determine nitrogen balance, electrolyte levels, and vitamin deficiencies can provide valuable insights into the patient’s nutritional status. Monitoring of blood glucose is paramount with adequate insulin as needed to maintain blood glucose levels between 140 and 180 mg/dL [14]. If patients exhibit signs or symptoms of acute or chronic malnutrition, further evaluation is warranted.

One of the more challenging aspects of NSTIs is assessing the patient’s calorie and protein needs as limited specific literature exists to aid in this process. The goal of nutrition support is to provide adequate calories and protein to blunt hypermetabolism, support immune function, and promote wound healing [14].

### 4.2. Energy Expenditure

Metabolic expenditure (and, subsequently, caloric needs) can be calculated using predictive formulas such as the Harris–Benedict and Mifflin–St. Jeor equations. Predictive formulas use anthropometric measurements such as height and weight as well as parameters such as age to estimate the basal metabolic rate or resting energy expenditure, which can then be used to estimate caloric requirements. Critically ill patients, including those with NSTIs, require a significantly increased caloric intake. For example, patients with severe stress may need 25–30 kcal/kg/day or more to meet their hypermetabolic demands. Because anthropometric measurements are not very accurate in critically ill patients due to the impact of the stress response on body composition, indirect calorimetry is considered the gold standard for evaluating energy expenditure in ICU patients [14,15,29].

The accuracy of weight-based predictive equations is further limited when assessing the nutritional needs of individuals with NSTIs who reside in larger bodies. Historical research by Majeksi and Alexander suggested that providing twice the basal energy needs for patients with NSTIs was associated with reduced complications [30]. However, a 2005 retrospective study by Graves et al. demonstrated, via the use of indirect calorimetry, that patients’ daily resting energy expenditure ranged from 60% to 199% of the basal energy expenditure assessed with the Harris–Benedict equation [29]. The authors concluded that, when indirect calorimetry cannot be performed, providing 25 kcals/kg of actual body weight per day or about 124% of basal energy expenditure would meet the needs of most NSTI patients [29]. This recommendation aligns with guidance from ASPEN and ESPEN [14,15]. However, due to the significant variability between patients, as well as the dynamic nature of metabolism over the course of critical illness, indirect calorimetry remains the optimal method to determine energy needs in NSTI patients [14,15,29].

### 4.3. Estimating Protein Needs

Determining the appropriate protein provision has long been a challenge in the critically ill population. When calculating macronutrient ratios, including adequate protein supplementation is crucial for NSTI patients to prevent further muscle wasting or protein loss and promote surgical wound healing. This is especially true if renal replacement therapy is required.

Protein recommendations range from 1.2 to 2.5 g/kg/day depending on the NSTI severity, with adjustments for CRRT and higher BMIs (Table 5) [14]. Randomized control trials and meta-analyses comparing higher and lower protein replacement amounts have not demonstrated clinical benefits with higher protein supplementation in the first 7–10 days of critical illness [31,32,33]. In patients with acute kidney injury, higher-protein regimens may be associated with worse clinical outcomes; however, if renal replacement therapy is required at any point during hospitalization, higher protein replacement is recommended [34]. In contrast, a meta-analysis of studies investigating higher- versus lower-protein interventions after 7–10 days of critical illness demonstrated a beneficial effect on short-term mortality and lean mass retention with higher protein replacement [35]. The strength in these studies is their support for protein supplementation in critically ill patients early after illness, and protein replacement should increase after the first week of hospitalization.

The limitation of these studies is their lack of research on protein requirements in the NSTI population. Therefore, it is appropriate to begin at the lower end of the recommended range of protein replacement and slowly increase the regimen based on factors such as wound healing and granulation, the need for renal replacement therapy, and promoting the sparing of skeletal muscle.

The widely available tools to individualize protein targets include the nitrogen balance and urea-to-creatinine ratio [36,37]. The nitrogen balance compares the nitrogen output in urine, which is a proxy for protein metabolism, to the protein intake from nutrition support [36]. A negative balance indicates inadequate protein intake to replace tissue breakdown, while a positive balance suggests tissue accretion through wound healing, muscle anabolism, or adipose tissue deposition. Equilibrium is defined as −2 to +2 g of nitrogen, per day [36]. Nitrogen balance studies will be inaccurate in individuals on renal replacement therapy, with significant serosanguinous losses, or with significant changes in their serum urea nitrogen [37]. The urea-to-creatinine ratio in critically ill patients can provide insight into the severity of inflammation, with elevated ratios indicating anabolic resistance and increased urea cycle metabolism [37].

## 5. Nutrition Interventions

### 5.1. Oral Diets

Patients who do not require mechanical ventilation should be initiated on oral diets unless dysphagia is suspected. Oral intake is often poor, related to anorexia from anesthesia, the stress response, taste changes from medications, and patient concerns regarding the location of wounds [38]. Protein intake, especially, presents a challenge to many hospitalized patients experiencing reduced appetite and taste changes. Diet restrictions should be limited to only those necessary. Oral nutrition supplements are associated with a reduced risk of developing malnutrition and should be incorporated as soon as inadequate intake is suspected [39]. Additionally, patients with frequent operative needs may benefit from protocols to minimize nil per os (NPO) status. Supplemental enteral nutrition may be required if patients cannot meet 60% of their nutrition needs through oral intake and can be provided as nocturnal cyclic feeds or post-meal boluses to encourage appetite [14].

### 5.2. Enteral Nutrition

Early initiation of enteral nutrition (EN) support is recommended, ideally within 24–48 h of admission to the intensive care unit, in patients who cannot receive oral diets [15,38]. In one study investigating patients with NSTIs, early EN was associated with fewer ventilator days, fewer hospital-acquired infections, and shorter lengths of stay [40]. Due to the benefit of early EN, initiation should not be delayed in an attempt to gain post-pyloric access unless gastric feeding is contraindicated or attempts at gastric feeding are unsuccessful [14]. However, early full nutrition can be harmful during the initial hypometabolic phase, and gradual advancement to full nutrition is recommended [11,41]. In patients presenting with septic shock and vasopressor-dependent shock, trophic EN over the first week of critical care has been shown to be non-inferior to goal EN [14,42]. Over the first week of critical care, an average delivery of 50–65% of assessed nutrition needs is adequate, with targets increased to greater than 80% of needs after the first week [14,43]. Reduced fasting protocols are associated with improved achievement of nutrition targets [44]. Volume-based feeding, where EN is ordered as a daily volume target and the hourly rate is adjusted to compensate for interruptions in infusion, improves nutrition intake but may not be adequate to compensate for the frequent and significant infusion interruptions experienced by patients with NSTIs [14].

In patients with vasopressor-dependent shock, early EN initiation is recommended and is associated with clinical benefit at norepinephrine doses of ≤0.3 mcg/kg/min [45]. Higher vasoactive support is associated with a lack of benefit from early EN, and one multi-center trial demonstrated an increased rate of non-occlusive bowel ischemia in patients randomized to early EN compared to those who received early parenteral nutrition support when norepinephrine doses exceeded 0.5 mcg/kg/min [46]. Although this study has limitations, its strength establishes the guidance that EN or parenteral nutrition is indicated for patients on vasopressors, and the choice between the two is largely based on the degree of vasopressor support. Routine monitoring of gastric residual volumes (GRVs) is no longer supported for most patients [14]. However, research demonstrates that eliminating GRV monitoring should exclude patients on vasoactive support, and GRV monitoring in patients receiving EN while on vasopressors may indicate early signs of intolerance. Increased gastric output, changes in stool output, and abdominal exams all require evaluation to determine whether a continuation of EN support is clinically appropriate [14].

Many enteral feeding formulas are available with modified macronutrient ratios and micronutrient contents to target specific goals, such as low-carbohydrate formulas for patients with diabetes and low-electrolyte formulas for patients with renal failure. Immunomodulators such as arginine are often avoided in patients presenting with septic shock [47]. Omega-3-fortified formulas favorably influence systemic inflammation without negatively impacting outcomes; however, additional research is needed on the role of immunomodulators in septic patients. High-protein-content formulas should be prioritized in patients with NSTI to support adequate nutrition intake for wound healing while reducing the reliance on modular protein supplements, such as whey or soy protein, that have lower rates of delivery [48].

### 5.3. Parenteral Nutrition

If enteral nutrition is contraindicated, parenteral nutrition (PN) support is recommended to meet nutrition needs and should be initiated within the first 5 days of admission in patients with high nutrition risk [49]. Initiation prior to hospital day 3 may not confer a benefit and may interfere with early metabolic processes, leading to a worsened nitrogen balance later during hospitalization [50]. Supplemental PN may be beneficial in patients who are unable to meet nutrition needs through EN due to intolerance, malabsorption, or poor oral intake and refusal of gastrointestinal access for supplemental EN [49].

Historically, PN has been associated with increased infectious complications, challenges with glycemic control, and hepatic dysfunction [51]. Many of these complications are related to practices of intentionally overfeeding through the use of PN support. Current research has demonstrated that PN in the intensive care unit prescribed with modern assessment standards is not associated with an increased risk of infection [51,52]. Similarly, hepatic dysfunction is primarily associated with overfeeding or prolonged PN dependence [51]. Cycling PN to infuse over less than 24 h, reducing the soybean oil content in lipid emulsions, considering trophic EN or small amounts of food by mouth, and avoiding overfeeding are strategies to mitigate the historical negative effects associated with PN support [51,52].

## 6. Nutrition Monitoring After Discharge

Nutrition intake at the time of transfer from the intensive care unit is often inadequate [11,38]. This gap continues after hospital discharge and requires ongoing monitoring of nutrition intake, regardless of the route of delivery, at all outpatient follow-up appointments until the nutrition status stabilizes and any nutritional decline experienced during hospitalization has been corrected [38]. The incorporation of nutrition services into post-discharge care is associated with better outcomes, reduced readmission rates, and improved quality of life; therefore, this should be considered as a component of post-discharge care planning [38]. Unfortunately, the availability of nutrition professionals in post-surgical clinics is limited, and community resources do not have broad experience with patients recovering from an NSTI.

### Ensuring Nutritional Adequacy

Close monitoring is essential to avoid complications of overfeeding (e.g., hyperglycemia, hepatic steatosis) or underfeeding (e.g., muscle wasting, poor wound healing). Monitoring patients’ nitrogen balance, caloric intake, and weight changes helps ensure that they are receiving adequate nutritional support. Regular assessments of energy expenditure and body composition using tools such as indirect calorimetry and bioelectrical impedance can also help refine nutritional interventions.

## 7. Summary

The management of necrotizing soft tissue infections requires comprehensive nutritional support, at times like the approach used in burn patients. NSTI patients experience heightened metabolic demands, and the goal of nutritional therapy is to support wound healing, immune function, and overall recovery. Nutritional needs should be assessed early after diagnosis of the NSTI, with enteral nutrition being the preferred route. Close monitoring is essential to avoid the complications of underfeeding or overfeeding, and nutritional adequacy can be evaluated through improvements in wound healing, weight maintenance, and immune function. With integrated, individualized, goal-directed nutritional support from the ICU to outpatient care, patients with NSTIs can achieve better long-term outcomes and recovery.

## Figures and Tables

**Table 1 jcm-14-03167-t001:** Types of necrotizing soft tissue infections.

Type	Incidence	Etiology	Patient Population	Affected Areas	Typical Microorganisms
**Type I**	70–80%	Polymicrobial	Elderly Medical comorbidities	Trunk Abdomen Perineum	Gram-positive cocci: non-group A streptococcus Gram-negative rods: enterobacteriaceae (*escherichia coli, enterobacter, klebsiella, proteus*) Anaerobes: *bacteroides, clostridium, peptostreptococcus, fusobacterium*
**Type II**	20–30%	Monomicrobial	Recent trauma or surgery IVDA	Extremities	Gram-positive cocci: group A beta-hemolytic streptococci, other beta-hemolytic streptococci, *staphylococcus aureus* (MRSA)
**Type III**	<1%	Aquatic microorganisms	Minor trauma Fresh and/or seawater exposure	Extremities	Gram-negative rods: *vibrio vulnificus, aeromonas hydrophila*
**Type IV**	<1%	Fungal	Immunocompromised patients Trauma/burns	Variable	Fungi: candida spp., *zygomycetes*

**Table 2 jcm-14-03167-t002:** Metabolic response to stress and injury.

Phase	Timing	Metabolic Activity	Characteristics	Nutritional Needs
**Early Acute Phase**	0–2 days	Hypometabolic	Decreased energy consumption Diminished mitochondrial capacity Hemodynamic instability	Early, hypocaloric nutrition Oral, enteral, or parenteral nutrition Assessment of protein calories
**Late Acute Phase**	2–7 days	Catabolic	Increased energy expenditure Muscle proteolysis Rapid tissue breakdown	Isocaloric nutrition Adequate protein intake Transition to oral or enteral nutrition
**Recovery Phase**	3–8 days	Anabolic Homeostatic	Resolution of inflammation	Nutritional rehabilitation Oral or enteral nutrition
**PICS ^‡^**	10–14 days	Anabolic resistance Catabolic	Smoldering inflammation Infectious recidivism Exercise intolerance	Hypercaloric nutrition Increased protein intake Incorporation of nutrition services at follow-up

^‡^ PICS: Persistent Inflammation, Immunosuppression, and Catabolism Syndrome.

**Table 3 jcm-14-03167-t003:** Malnutrition assessments.

	Subjective Global Assessment (SGA)	Academy of Nutrition and Dietetics and ASPEN Indicators of Malnutrition (AAIM)	Global Leadership Initiative on Malnutrition (GLIM)
**Year developed**	1982	2011	2018
**Settings and application**	Clinical setting Adult patients	Clinical setting Adult patients	All settings Adult patients
**Criteria and parameters**	**History** Weight changes Dietary intake Gastrointestinal symptoms Changes in functional capacity	**History** Energy intake Interpretation of weight loss	**Etiologic criteria** Reduced food intake or assimilation Inflammation and disease burden
	**Physical examination** Subcutaneous fat Muscle wasting Edema Ascites	**Physical examination** Body fat (loss) Muscle mass (loss) Fluid accumulation Reduced grip strength	**Phenotypic criteria** Unintentional weight loss Low body mass index Reduced muscle mass
**Scoring and diagnosis**	A: Well nourished B: Moderately malnourished C: Severely malnourished	Non-severe (moderate) malnutrition Severe malnutrition	Stage 1: Moderate malnutrition Stage 2: Severe malnutrition
**Benefits**	Gold standard for malnutrition assessment Simple Noninvasive	Structured Validated	Global application Objective Standardized Comprehensive
**Limitations**	Subjective	Less sensitive and specific compared to GLIM	Complex

**Table 4 jcm-14-03167-t004:** Initial nutritional assessment.

History	Physical Examination	Laboratory Studies	Imaging Studies
Medical comorbidities	Muscle wasting	Serum proteins	DEXA
Recent trauma, illness, or surgery	Body habitus	Acute-phase reactants	Computed tomography
Recent weight loss	Skinfold thickness	Micronutrient levels	Magnetic resonance
Changes in functional status	Skin, hair, or nail abnormalities	Nitrogen balance	Ultrasound
Psychiatric conditions	Edema or anasarca	Functional tests	Bioelectrical impedance

**Table 5 jcm-14-03167-t005:** Summary of protein recommendations for critically ill patients with and without renal replacement therapy.

BMI Range	Initial Protein	Maximum Protein	CRRT
**<18.5**	1.5 g/kg ABW	2 g/kg ABW	2–2.5 g/kg ABW
**18.5–24.9**	1.5 g/kg ABW	2 g/kg ABW	2–2.5 g/kg ABW
**25–29.9**	1.5 g/kg ABW	2 g/kg ABW	2–2.5 g/kg ABW
**30–34.9**	2 g/kg/IBW	2.5 g/kg/IBW	Consider 2.5–3 g/kg/IBW ^‡^
**35–39.9**	2 g/kg/IBW	3 g/kg/IBW	Consider 2.5–3 g/kg/IBW
**≥40**	2 g/kg/IBW	3 g/kg/IBW	Consider 2.5–3 g/kg/IBW

^‡^ No recommendation available. Consider adjusting protein provision based upon the patient’s response to therapy. ABW: actual body weight; BMI: body mass index; IBW: ideal body weight.

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
