# Peer review of "Nutritional Support for Necrotizing Soft Tissue Infection Patients: From ICU to Outpatient Care"

_jcm, 2025, doi:10.3390/jcm14093167_

Round 1

Reviewer 1 Report

Comments and Suggestions for Authors

The topic of nutritional support is crucial  yet often underrated, particularly in complex patients with necrotizing soft tissue infections.

The authors provide a comprehensive overview of the diagnostic and management strategies related to this complex issue.

Their discussion of the distinct phases of the metabolic response, along with the emphasis on individualized nutritional support throughout the recovery process, is both insightful and clinically relevant. The review clearly outlines the criteria for diagnosing malnutrition and addresses the adjustment of caloric and protein needs based on comorbid factors such as renal failure and the use of continuous renal replacement therapy (CRRT). Furthermore, the authors effectively describe various nutritional interventions and present clear guidelines for initiating enteral and parenteral nutrition.

Overall, this is a well-structured and informative review article that will be of significant value to clinicians involved in critical care, nutritional management, and postoperative recovery of necrotizing soft tissue infection patients.

Author Response

Comment 1: Overall, this is a well-structured and informative review article that will be of significant value to clinicians involved in critical care, nutritional management, and postoperative recovery of necrotizing soft tissue infection patients.

Response 1: Thank you for taking the time to review this manuscript.

Reviewer 2 Report

Comments and Suggestions for Authors

This is a well-structured and comprehensive review of nutritional management in patients with necrotizing soft tissue infections (NSTIs), offering a valuable synthesis of current evidence across the entire continuum of care—from ICU to outpatient recovery. The manuscript effectively highlights the distinct metabolic response in NSTIs and provides practical guidance on assessment tools, energy and protein needs, and feeding strategies. The inclusion of the PICS phase and emphasis on post-discharge nutrition are timely and relevant.

To further improve the manuscript, the authors may consider adding brief critical commentary on the strength or limitations of key cited studies, especially where evidence remains inconclusive (e.g., protein dosing in acute illness or early enteral vs. parenteral nutrition). Some transitions between sections could also be smoother, and minor grammatical polishing would enhance readability. Finally, a visual summary or algorithm outlining nutrition interventions across the recovery phases would strengthen the practical applicability of this work.

Author Response

Comment 1: To further improve the manuscript, the authors may consider adding brief critical commentary on the strength or limitations of key cited studies, especially where evidence remains inconclusive (e.g., protein dosing in acute illness or early enteral vs. parenteral nutrition). Some transitions between sections could also be smoother, and minor grammatical polishing would enhance readability. Finally, a visual summary or algorithm outlining nutrition interventions across the recovery phases would strengthen the practical applicability of this work.

Response 1:  Thank you for taking the time to review this manuscript. Critical commentary was provided for key studies where evidence remains inclusive. In addition, Table 3 was updated to include limitations of key assessments. In regard to the request for an algorithm outlining nutrition interventions, Table 2 was updated to include nutrition interventions across recovery phases.  Lastly, transitions between sections were improved.

To the editors: some of the tables were cut off, we expanded their footprints to include the entire table.  Please note this if additional changes are made to the manuscript.